# Clinical, Meteorological, and Air Quality Factors Associated with Ambulatory Pediatric Respiratory Syncytial Virus Infection in Machala, Ecuador, 2018–2023

**DOI:** 10.3390/ijerph22020272

**Published:** 2025-02-13

**Authors:** Manika Suryadevara, Dongliang Wang, Freddy Pizarro Fajardo, Jorge-Louis Carrillo Aponte, Froilan Heras, Cinthya Cueva Aponte, Irene Torres, Joseph B. Domachowske

**Affiliations:** 1Department of Pediatrics, SUNY Upstate Medical University, 750 East Adams Street, Syracuse, NY 13210, USA; wangd@upstate.edu (D.W.); domachoj@upstate.edu (J.B.D.); 2Hospital General del Norte de Guayaquil IESS Los Ceibos, Guayaquil 090615, Ecuador; fredduos@hotmail.com; 3Research Center at Hospital Teofilo Davila, SUNY Upstate Medical University, Machala 070206, Ecuador; jorge.carrillo@ucuenca.edu.ec (J.-L.C.A.); herasfroilan@gmail.com (F.H.); cin_ka10@hotmail.com (C.C.A.); 4Fundacion Octaedro, Quito 170135, Ecuador; irene.torres@octaedro.edu.ec

**Keywords:** pediatrics, respiratory syncytial virus, respiratory virus, Ecuador, tropics

## Abstract

Implementation of RSV prevention strategies requires an understanding of seasonal virus epidemiology; yet, such data are lacking in many tropical regions. We describe the seasonality and environmental factors associated with RSV activity in Machala, Ecuador. From July 2018 to July 2023, we analyzed nasopharyngeal samples from children younger than 5 years with an acute respiratory illness using Biofire FilmArray v.1.7™. Meteorological data were obtained from Ecuador’s Instituto Nacional de Meteorología e Hidrología. RSV detection was associated with mean outdoor air temperature (threshold 27 °C, area under the curve (AUC) 0.74, *p* < 0.001) that was even stronger when using a 4-week lag (threshold 27 °C, AUC 0.865, *p* < 0.001) and with precipitation greater than 0.5 mm/week using a 6-week lag (*p* = 0.008). This finding could provide guidance for the ideal timing to improve RSV surveillance and implement RSV prevention measures in Machala, including universal administration of extended half-life monoclonal antibody to infants entering their first RSV season.

## 1. Introduction

Respiratory syncytial virus (RSV) is a leading cause of acute respiratory tract infection (ARTI) among children younger than 5 years of age. In 2019, an estimated 33 million lower respiratory tract infections (LRTI), 3.6 million hospitalizations, and 26,300 deaths were associated with RSV globally, with substantial disease burden on infants younger than 6 months residing in low- and middle-income countries (LMICs) [1]. While safe and effective RSV treatment beyond supportive care remains elusive, recent successes aimed at infant RSV prevention are now being implemented worldwide.

Current available pharmacologic strategies for pediatric RSV infection prevention include maternal RSV immunization and/or the administration of extended half-life monoclonal RSV antibody to infants just prior to and during RSV season. Palivizumab, administered as monthly injections during peak RSV season, was used to prevent RSV in young infants; however, cost and logistical concerns limited its use and availability to only the highest-risk infants living in developed countries of the world [2]. The recent licensure and approval of the first extended half-life monoclonal antibody, nirsevimab, has broadened the feasibility of incorporating this strategy for universal infant protection across all health care systems, including those identified as LMICs [2].

Implementing public health strategies to prevent RSV disease burden among infants and young children requires a detailed understanding of regional, seasonal virus epidemiology, healthcare resources and utilization, and factors that may influence the timing or severity of disease [2,3]. Unlike temperate regions of the world, where RSV activity shows clear seasonal patterns during the colder winter months, reports from tropical areas indicate less predictable patterns of RSV activity, with some regions describing year-round RSV activity [4,5]. Periods of increased RSV disease burden have been associated with air quality and meteorologic factors in some regions but not in others [4,6,7,8,9,10,11,12,13,14,15,16,17,18,19]. As such, it is impossible to generalize patterns of respiratory virus epidemiology across tropical regions, even within the same country, without knowledge of regional epidemiology. Understanding epidemiology is important for different infectious agents in the various regions across the world, as has been previously described for both respiratory and gastrointestinal pathogens [20,21,22,23].

Ecuador is a middle-income country located on the Pacific coast of South America. Traversed by the equator, Ecuador has a total area of 283,560 km^2^ and an estimated total population of 17,374,000. Within the country, elevations range from sea level at the coast to over 6000 m in the Andes Mountains. The coastal city of Machala, with a population of 231,260 (3000 residents per square mile) is positioned at 4° S latitude just a few meters above sea level. The rainy season in Machala typically occurs between January and April, with the driest months occurring between June and August.

In 2019, national data from Ecuador indicated that LRTIs were among the top four causes of death and the leading cause of communicable disease among all age groups combined [24,25]. Despite the clear public health burden associated with ARTIs, detailed surveillance describing virus-specific epidemiology from the country are lacking. Here, we report and compare the clinical characteristics, temporal patterns, meteorologic factors, and air quality parameters that are associated with RSV activity among Ecuadorian children living in the coastal city of Machala before and after the onset of the SARS-CoV-2 pandemic.

## 2. Materials and Methods

This 5-year cross-sectional surveillance study of medically attended outpatient ARTIs among Ecuadorean children less than 5 years of age took place between July 2018 and July 2023. Enrollment was temporarily halted in March 2020 due to restrictions imposed in response to the SARS-CoV-2 pandemic and re-started on 5 August 2020. Data obtained prior to the pause in enrolment are referred to as ‘before the onset of the SARS-CoV-2 pandemic’, while data obtained after enrollment re-started in August 2020 are referred to as ‘after the onset of the SARS-CoV-2 pandemic’. This project is approved by the State University of New York Upstate Medical University Institutional Review Board (IRB number 1102402) and by the Ministry of Health of Ecuador.

### 2.1. Subject Enrollment

Eligibility for inclusion and recruitment logistics were previously described [26]. Briefly, children younger than 5 years of age presenting to a study-designated ambulatory clinic in Machala, Ecuador with a suspected ARTI were eligible for enrollment. An ARTI was defined by the presence or parental report of two or more of the following symptoms for fewer than 8 days: temperature ≥ 38 °C, nasal congestion or discharge, cough, tachypnea, wheezing, rales, hypoxia, or apnea. Children in foster care, those with parents unable or unwilling to provide consent, and those hospitalized or treated with antibiotics within 30 days prior to enrollment were excluded from study participation.

After completing the informed consent process, demographic and clinical data were recorded. For study purposes, upper respiratory tract infections (URTI) included the primary diagnoses of nasopharyngitis and laryngotracheitis, while LRTI included diagnoses of bronchitis, bronchiolitis, and pneumonia.

A nasopharyngeal swab sample was obtained, processed and tested for the presence of respiratory pathogen nucleic acids (adenoviruses, coronaviruses HKU1, NL63, 229E, OC43, human metapneumovirus, human rhinoviruses/enteroviruses, influenza viruses A-H1N1, A-H3N2, B, parainfluenza viruses 1, 2, 3, 4, respiratory syncytial virus, *Bordetella pertussis*, *Chlamydophila pneumoniae*, and *Mycoplasma pneumoniae*) using the multiplex PCR-based BioFire Film Array™ Respiratory Panel v1.7 (BioFire Diagnostics, LLC, Salt Lake City, UT, USA). Results reported herein as “RSV not detected” represent those samples testing negative for the listed pathogens and those testing positive for any pathogen other than RSV.

### 2.2. Meteorologic Data Collection

Meteorologic parameters were obtained from data downloaded from weather towers maintained by Ecuador’s Instituto Nacional de Meteorología e Hidrología. Weather tower instruments were calibrated every 3 months and immediately following any necessary repairs. Meteorologic parameters, including temperature (°C), relative humidity (%), barometric pressure (W/m^2^), and precipitation (mm) were captured each minute, on a continuous basis and summarized as daily, weekly, and/or monthly averages. Means, medians, and ranges were calculated for each week. From the measured data, dew points were calculated using the following formula:B = (ln (RH/100) + ((17.27 × T)/(237.3 + T)))/17.27D = (237.3 × B)/(1 − B)
where T = air temperature (°C); RH = relative humidity (%); B = intermediate value (no units); and D = dewpoint (°C).

### 2.3. Air Quality Data Collection

Air quality data were collected up to three times daily using Aeroqual series 200 monitors with calibrated sensor heads that detect PM 10 (particulate matter ≤ 10 microns in diameter), PM 2.5 (particulate matter ≤ 2.5 microns in diameter, “fine particles”) [range of detection 0.001–1.000 mg/m^3^; calibration accuracy of <±0.5 ppm for 0 to 5 ppm, and <±10% for 5–100 ppm)], carbon monoxide [range of detection 0–25 ppm; calibration accuracy of <±0.5 ppm for 0 to 5 ppm, and <±10% for 5–25 ppm], and nitrogen dioxide [range of detection 0 to 1 ppm; calibration accuracy of <±0.02 ppm for 0 to 0.2 ppm and <±10% for 0.2 to 1 ppm]. Aeroqual instruments were calibrated every 3 months and immediately following any necessary repairs. Measurements for each analyte were summarized and reported as weekly means.

### 2.4. Statistical Analysis

Descriptive statistics were used. Continuous data were compared by *t*-test. Categorical data were compared using the Fisher exact or chi-square test, as appropriate. Significance was set a priori to a *p* value of <0.05. Receiver operating characteristics (ROC) analyses were performed to evaluate the performance of the potential classification factors/models by calculating the sensitivities (true positive rates) and specificities (true negative rates) at each classification threshold. The area under the ROC curve represents an overall measure of the classification accuracy, and a bootstrap based test was performed to assess whether the AUC was different from 0.5, the AUC of a noninformative classification model. The optimal threshold was chosen to maximize the sum of sensitivity and specificity. All factors with significant differences between the two groups were included in multiple logistic regression to derive a composite score for classification, as a weighted average of individual factors. LASSO penalty was used as a penalty term of model complexity to handle correlations between test scores. The differences between different logistic models were tested by Rao’s score test.

## 3. Results

### 3.1. All Enrolled Subjects

During the study period, 1251 subjects (695, 56% males) were enrolled (Table 1). The mean age was 19 months, and 594 (47%) subjects were less than 1 year of age. Of the 1251 nasopharyngeal samples collected, 873 (70%) tested positive for the detection of at least one pathogen on the multiplex PCR respiratory panel, including 114 (9%) samples that were positive for RSV detection. Nasal congestion (1186, 95%) and fever (1017, 81%) were the most commonly reported symptoms among all enrolled subjects.

Subjects whose nasopharyngeal samples were positive for RSV detection were more likely than those whose samples detected another pathogen to: report wheezing (24/114 (21%) vs. 51/759 (7%), *p* < 0.001), have a longer mean duration of symptoms (3.6 vs. 3.2 days, *p* = 0.02), be diagnosed with a LRTI (28/144 (25%) vs. 80/759 (11%), *p* < 0.001), and be treated with antibiotics (30/114 (26%) vs. 99/759 (13%), *p* < 0.001) (Table 1). No differences were noted between the two groups with regards to age, gender, or an age-appropriate history of having received an influenza or pertussis vaccine.

### 3.2. Subjects Whose Nasopharyngeal Samples Were Positive for the Detection of RSV

The nasopharyngeal samples from 114 subjects were positive for RSV detection. The mean age of subjects with documented RSV infection was 18 months (range of 1–59 months). A total of 57 (50%) of the subjects who tested positive for RSV were younger than 1 year of age.

In addition to the detection of RSV, 41 (36%) nasopharyngeal samples also tested positive for a second pathogen (24 rhinovirus/enterovirus, 6 adenovirus, 4 parainfluenza virus, 4 human coronavirus, 1 human metapneumovirus, 1 influenza virus, and 1 *Chlamydophila pneumoniae*), and 2 (1.8 %) tested positive for 2 other pathogens (1 with rhinovirus/enterovirus and adenovirus, 1 with rhinovirus/enterovirus and parainfluenza virus). The characteristics of the subjects whose nasopharyngeal sample detected RSV alone and those detecting RSV and another pathogen are shown in Table 2. There were no statistical differences between these two groups.

Subjects 1 year of age and older were more likely to report fever with their ARTI compared to those less than 1 year of age (53 (93%) vs. 43 (75%), *p* = 0.01) and were more likely to have received vaccines against influenza (45 (75%) vs. 15 (26%), *p* < 0.001) and pertussis (23 (40%) vs. 17 (30%), *p* < 0.001). There were no other statistical associations between subject age, presenting symptoms or visit diagnosis.

Among the 114 nasopharyngeal samples that tested positive for the detection of RSV, 38 (33.3%) were collected before the onset of the SARS-CoV-2 pandemic (Table 1). Compared with subjects enrolled prior to the onset of the pandemic, those enrolled after pandemic onset were more likely to present with fever (71 (93%) vs. 25 (66%), *p* = 0.02) and less likely to present with cough (36 (47%) vs. 36 (95%), *p* < 0.001) or wheeze (8 (11%) vs. 16 (42%), *p* < 0.001). Similarly, subjects with RSV infection enrolled after pandemic onset were more likely to be diagnosed with an URTI (69 (91%) vs. 16 (42%), *p* < 0.001) and less likely to be diagnosed with an LRTI (7 (9%) vs. 21 (55%), *p* < 0.001). Subjects enrolled after the pandemic onset who tested positive for RSV detection were also older (19.6 months) than those who were identified prior (14.4 months), although the trend was not found to be statistically different (*p* = 0.09).

### 3.3. RSV Seasonality

Of the 1251 nasopharyngeal samples tested during 221 weeks of the study when samples were collected, the 114 RSV-positive samples were collected over 55 (25%) different calendar weeks. 

Prior to the SARS-CoV-2 pandemic, RSV activity was clustered between February and May (2019) and February and March (2020), when study enrollment was halted. After pandemic onset (2022, 2023), RSV activity was detected more frequently during April, May, and June, although the seasonal trends were not found to be statistically different (Figure 1).

### 3.4. Meteorologic Data

Outdoor air temperature showed seasonal variation over the course of the study with peak temperatures occurring during March (mean 28 °C, maximum 34 °C) and low temperatures observed during August (mean 23 °C, minimum 19.6 °C). Mean relative humidity gradually increased over the course of the calendar year between 2018 and 2023, from a low of 71% (range 8–100%) in January to a high of 85% (range 10–100%) in November. Mean dew points gradually increased from a low of 21 °C (range 19–23 °C) in September to a high of 24 °C (range 22–25 °C) in March. Mean barometric pressure consistently measured between 1010 W/m^2^ and 1012 W /m^2^.

RSV detection events were associated with periods of higher mean air temperature and dew points, and lower relative humidity and barometric pressure (Table 3). Mean temperature, however, remained the only factor consistently associated with RSV detection after both univariate and multivariate logistic regression analyses were performed (Figure 2A, Table 4). The association between mean air temperature and RSV detection events was even stronger when analyzed using a 4-week lag, with a temperature threshold of 27 °C (AUC 0.865, *p* < 0.001) (Figure 2B). RSV detection was also associated with rainfall greater than 0.5 mm per week, with a 6-week lag (*p* = 0.008).

### 3.5. Air Quality Data

Monthly mean PM 2.5 concentration was lowest in February and March at 5.8 mcg/m^3^ (range 3–9.67 mcg/m^3^) and highest in September at 12.5 mcg/m^3^ (range 7–21.3 mcg/m^3^). Monthly mean NO_2_ concentrations varied throughout the year, ranging from 0.00332 ppm to 0.0563 ppm. Monthly mean CO concentrations clustered between 0.1384 ppm and 0.2046 ppm between August and December, while remaining below 0.036 ppm during the rest of the year.

Of the air quality analytes surveyed, PM 2.5 concentrations were positively associated with RSV detection, while CO concentrations were negatively associated with RSV detection events when the nonparametric Wilcoxon rank sum test was performed to assess differences in medians (Table 3). Multivariate logistic regression analysis, however, did not support these associations. PM 10 and NO_2_ concentrations were not associated with RSV detection events.

## 4. Discussion

Over the five-year study period, RSV was detected in 9% of participants, with predictable seasonal patterns identified. Mean temperature was the factor most closely associated with RSV detection. Specifically, we observed that an increase in mean outdoor air temperature above 27 °C predicted RSV detection events, with the highest associations occurring after a four-week lag period. Dewpoint calculations also trended towards significance using multivariate analysis; however, ambient temperature is included in this calculation and likely plays a primary role in the observed association. Stable high temperatures in tropical regions may facilitate RSV survival in large-particle aerosols, particularly during rainy seasons, allowing for increased transmission during these weather patterns [27].

Participants with nasopharyngeal samples that tested positive for the detection of RSV who were enrolled after the onset of the SARS-CoV-2 pandemic were older, more likely to present with a fever, and less likely to be diagnosed with an LRTI than those enrolled prior to the pandemic. This finding, consistent with data published from other areas of the world, is likely the result of a delay in population-based first-time RSV infections as a result of non-pharmaceutical interventions put into place to limit SARS-CoV-2 transmission during the pandemic [28]. RSV circulated at very low levels during national lockdowns imposed in early 2020. As policies and practices led to less stringent use of the non-pharmaceutical interventions to limit SARS-CoV-2 transmission, a notable increase in out-of-season RSV activity occurred across the globe, with increased detection rates among children older than 1 year of age reported among both inpatient and outpatient cohorts [28,29,30,31,32].

We found that in the hot semi-arid climate of Machala, Ecuador, RSV was detected most frequently during the periods of highest rainfall (February to April), with no cases of RSV detected during the driest months (August and September). This observation is consistent with previously published reports of RSV infection in Ecuador [33,34,35]. Worldwide, rainfall has a variable impact on RSV infection, with a positive association reported in Scotland, Korea, and Thailand, and a negative association reported in Malaysia [14,15,36,37,38]. Meteorological factors influence respiratory virus activity through promoting virus survival, stability, replication, and transmissibility.

It is important to note that, in Machala, the periods of highest rainfall are also the periods associated with the warmest air temperatures. In our study, the mean air temperature was the only meteorological or air quality variable to remain positively associated with RSV detection, even after multivariate regression, a finding that is contrary to RSV epidemiology in temperate climates [39]. RSV incidence in subtropical and temperate regions is positively correlated with lower ambient temperatures and higher relative humidity, while in the tropical regions, the correlations between RSV incidence and these factors are much more variable and inconsistent [40]. Taken together, these observations highlight some of the potential hazards of extrapolating epidemiologic data described from ‘similar’ regions or neighboring countries and emphasize the importance of collecting region-specific data to best guide implementation of RSV prevention strategies.

In Machala, RSV detection events were associated with increasing concentrations of PM 2.5. The positive association between increasing PM 2.5 concentrations and RSV detection and bronchiolitis severity have been described [41,42]. Nearly 25% of inhaled small particulate matter reaches the alveoli, where it has been found to increase oxidative stress of the epithelial cells and ultimately lead to inflammation of and injury to the lung parenchyma [43]. Identifying and eliminating sources of air pollution that contribute to high concentrations of fine particles in the air has the potential to reduce RSV disease incidence and severity. Surprisingly, we found that RSV detection events decreased with increasing concentrations of CO in the air. While not a consistent finding in all locations, the trend has also been reported in Colombia and Singapore [16,44]. It is hypothesized that the antimicrobial and anti-inflammatory properties of CO may have the potential to protect the host from some of the deleterious effects of RSV infection [45].

We acknowledge several limitations to our study design that limit the generalizability of our results. The sample size may be insufficient to detect other important associations between RSV detection events and weather or air quality. The climate and air quality observed in Machala likely differs substantially from other areas of Ecuador given the variety of ecological niches present across the nation.

## 5. Conclusions

In Machala, the mean air temperature was the single most influential factor for RSV detection events, with an increase in mean temperature over 27 °C highly predictive of a rise in RSV cases four weeks later. This finding could provide Ecuador’s Ministry of Health officials with guidance about the ideal timing to improve RSV surveillance and implement infant RSV prevention measures each RSV season, including the universal all-infant administration of extended half-life monoclonal RSV antibody.

## Figures and Tables

**Figure 1 ijerph-22-00272-f001:**
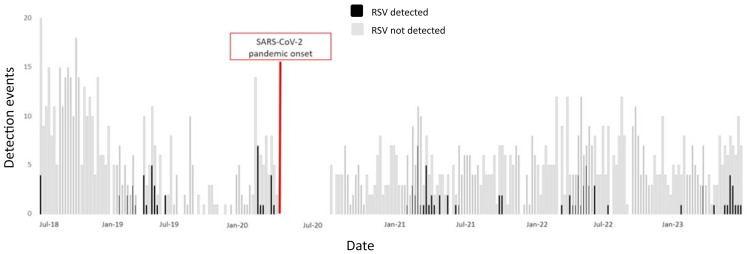
Shown is the study timeline (*x*-axis) from July 2018 through July 2023 indicating respiratory syncytial virus (RSV) detection events (*y*-axis) in black bars, superimposed onto detection events for any of the other pathogens tested using grey bars. The pause in study enrollment imposed by the SARS-CoV-2 pandemic is identified in red.

**Figure 2 ijerph-22-00272-f002:**
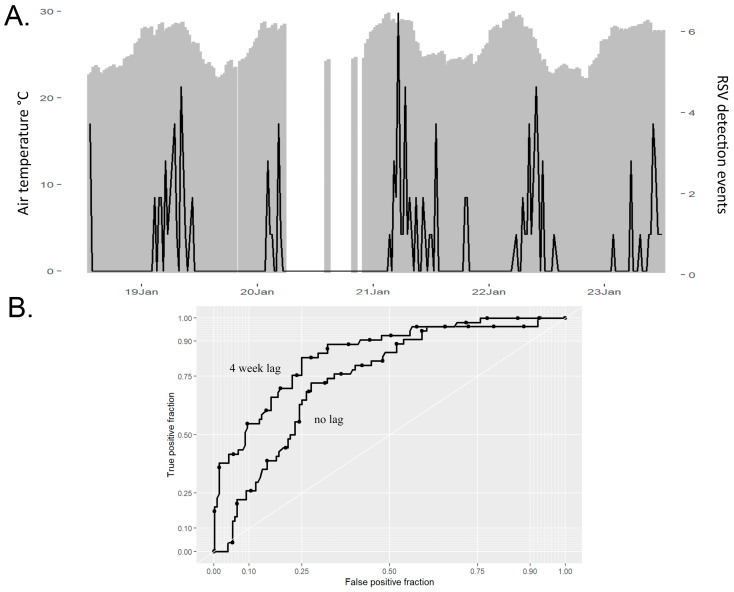
(**A**) Shown is the study timeline (*x*-axis) from July 2018 through July 2023 with respiratory syncytial virus (RSV) detection events (right *y*-axis) in black, superimposed onto weekly mean air temperatures in grey (left *y*-axis). (**B**) Shown are the receiver-operating characteristic (ROC) curves for mean air temperature associated with respiratory syncytial virus (RSV) detection events with and without imposing a 4-week lag from achieving the temperature threshold (27 °C).

**Table 1 ijerph-22-00272-t001:** Demographic and clinical illness characteristics of study participants seeking outpatient medical care for an acute respiratory illness in Machala, Ecuador.

		All Nasopharyngeal Samples			Nasopharyngeal Samples Detecting RSV †		
	Total	Pathogens Other than RSV Detected	RSV Detected	P*	Subjects Enrolled Prior to Pandemic Onset	Subjects Enrolled After Pandemic Onset	P**
Study participants	1251	759	114		38	76	
Mean age in months (range)	19, 1–59	19, 1–59	18, 1–59	0.44	14.4 (1–54)	19.6 (1–59)	0.09
Male gender, n (%)	695 (56)	412 (54)	63 (55)	0.84	23 (61)	40 (53)	0.4
Symptoms							
Fever, n (%)	1017 (81)	612 (81)	96 (84)	0.36	25 (66)	71 (93)	0.02
Nasal congestion, n (%)	1186 (95)	723 (95)	113 (99)	0.08	38 (100)	75 (99)	1
Cough, n (%)	641 (51)	401 (53)	72 (63)	0.05	36 (95)	36 (47)	<0.001
Wheeze, n (%)	93 (7)	51 (7)	24 (21)	<0.01	16 (42)	8 (11)	<0.001
Mean symptom duration in days (range)	3.3, 1–10	3.2, 1–7	3.6, 1–7	0.02	3.84 (2–7)	3.43 (1–7)	0.2
Upper respiratory tract infection, n (%)	1081 (86)	675 (89)	85 (75)	<0.01	16 (42)	69 (91)	<0.001
Nasopharyngitis, n (%)	1009 (81)	611 (81)	78 (68)	<0.01	10 (26)	68 (89)	
Laryngotracheitis, n (%)	68 (5)	18 (2)	1 (1)	0.5	1 (3)	0	
Lower respiratory tract infection, n (%)	124 (10)	80 (11)	28 (25)	<0.01	21 (55)	7 (9)	<0.001
Bronchiolitis, n (%)	100 (8)	62 (8)	21 (18)	<0.01	17 (45)	4 (5)	
Bronchitis, n (%)	8 (1)	7 (1)	1 (1)	1	1 (3)	0	
Pneumonia, n (%)	16 (1)	11 (1)	6 (5)	<0.01	3 (8)	3 (4)	
Treated with antibiotics, n (%)	176 (14)	99 (13)	30 (26)	<0.01	10 (26)	20 (26)	1
Influenza vaccine received, n (%)	646 (52)	395 (52)	60 (53)	0.91	21 (55)	39 (51)	0.7
Pertussis vaccine received, n (%)	1104 (88)	667 (88)	100 (88)	0.96	31 (82)	69 (91)	0.16

† RSV respiratory syncytial virus; *P represents the statistical difference between subjects whose nasopharyngeal samples tested positive for the detection of RSV and those whose samples tested positive for the detection of pathogens other than RSV. *T*-test was performed to compare the ages and symptom duration. Chi-square was performed when comparing categorical data. **P represents the statistical difference between subjects with RSV detected before and after the pandemic onset. Chi-square was performed to compare the categorical data. *T*-test was performed to compare ages and symptom duration.

**Table 2 ijerph-22-00272-t002:** Demographic and clinical illness characteristics of 114 RSV-positive study participants seeking outpatient medical care for an acute respiratory illness in Machala, Ecuador.

	RSV † Alone Detected	RSV Detected with Additional Pathogen	P*
Study participants	73 (64)	41 (36)	
Mean age in months (range)	17 (1–59)	19 (1–59)	0.67
Male gender, n (%)	38 (52)	25 (61)	0.36
Symptoms			
Fever, n (%)	63 (86)	33 (80)	0.41
Nasal congestion, n (%)	72 (99)	41 (100)	
Cough, n (%)	44 (60)	28 (68)	0.39
Wheeze, n (%)	15 (21)	9 (22)	0.86
Mean symptom duration in days (range)	3.5 (1–7)	3.6 (1–7)	0.75
Upper respiratory tract infection, n (%)	50 (68)	29 (71)	0.71
Nasopharyngitis, n (%)	49 (67)	29 (71)	
Laryngotracheitis, n (%)	1 (1)	0	
Lower respiratory tract infection, n (%)	16 (22)	11 (27)	0.71
Bronchiolitis, n (%)	11 (15)	10 (24)	
Bronchitis, n (%)	0	1 (2)	
Pneumonia, n (%)	5 (7)	0	

† RSV respiratory syncytial virus; *P represents the statistical difference between subjects whose nasopharyngeal samples tested positive for the detection of RSV and those whose samples tested positive for the detection of pathogens other than RSV. *T*-test was performed to compare the ages and symptom duration. Chi-square was performed when comparing categorical data.

**Table 3 ijerph-22-00272-t003:** Meteorologic and air quality factors associated with respiratory syncytial virus (RSV) detection in Machala, Ecuador.

Variable	RSV Detected	RSV Not Detected	AUC †	P*
Mean temperature				
n	54	152	0.74	<0.01
Mean (SD)	27.2 (1.6)	25.4 (2.1)		
Median (IQR)	27.7 (1.6)	24.8 (3.8)		
Min, Max	22.7, 28.9	22, 29.8		
Q1, Q3	26.7, 28.2	23.5, 27.4		
Mean dew point				
n	54	152	0.692	<0.01
Mean (SD)	22.5 (1.4)	21.5 (1.5)		
Median (IQR)	22.7 (1.6)	21.6 (2.6)		
Min, Max	19.5, 24.8	18.8, 24.9		
Q1, Q3	21.9, 23.5	20.1, 22.7		
Mean relative humidity				
n	51	130	0.616	0.016
Mean (SD)	75.4 (5)	77.7 (7.1)		
Median (IQR)	76 (7.4)	78.2 (11.1)		
Min, Max	64.6, 83.9	61.8, 92.2		
Q1, Q3	72.2, 79.6	72.1, 83.2		
Mean barometric pressure				
n	54	152	0.666	<0.01
Mean (SD)	1010.5 (1.2)	1011.2 (1.3)		
Median (IQR)	101.2 (1.5)	1011.4 (1.7)		
Min, Max	1007.7, 1013.1	1007.6, 1013.6		
Q1, Q3	1009.8, 1011.3	1010.4, 1012.1		
Particulate matter (PM) 2.5 mcg/m^3^				
n	55	166	0.606	0.02
Mean (SD)	8.56 (4.5)	9.49 (3.56)		
Median (IQR)	8.3 (3.9)	9.5 (4.6)		
Min, Max	3, 34.3	2.3, 21.3		
Q1, Q3	6.7, 11.3	6, 9.9		
Carbon monoxide (CO) ppm				
n	55	166	0.394	<0.01
Mean (SD)	0.01 (0.05)	0.11 (0.29)		
Median (IQR)	0 (0)	0 (0.07)		
Min, Max	0, 0.24	0, 1.73		
Q1, Q3	0, 0	0, 0.07		

† AUC is the area under the Receiver Operating Characteristics (ROC) curve to classify the dissimilatory capacity of the variable to assess whether RSV was detected. *P values to test whether the AUC is different from 0.5, the AUC of a non-informative classifier. The test is equivalent to the Wilcoxon rank sum test.

**Table 4 ijerph-22-00272-t004:** Odds ratios of respiratory syncytial virus (RSV) detection events associated with meteorologic and air quality factors using univariate and multivariate logistic regressions.

	Odds Ratio of RSV Detection Without Lag					
	Univariate Logistic Regression			Multivariate Logistic Regression		
	Odds Ratio	95% CI	*p*	Odds Ratio	95% CI	*p*
Weekly mean temperature	1.58	1.32, 1.91	<0.01	2.28	1.52, 3.74	<0.01
Weekly mean dew point	1.59	1.27, 2.02	<0.01	1.20	0.79, 1.78	0.37
Weekly mean relative humidity	0.95	0.90, 1.0	0.038	1.05	0.97, 1.16	0.25
Weekly mean barometric pressure	0.67	0.52, 0.85	<0.01	1.32	0.89, 1.97	0.17
Weekly mean PM 2.5 † (mcg/m^3^)	0.30	0.12, 0.67	<0.01	1.18	0.19, 1.37	<0.01
Weekly mean CO † (ppm)	0.93	0.85, 1.01	0.12	0.53	0.19, 1.37	0.21
	**Odds ratio of RSV detection with 4-week lag**					
	**Univariate logistic regression**			**Multivariate logistic regression**		
	**Odds ratio**	**95% CI**	* **p** *	**Odds ratio**	**95% CI**	* **p** *
Weekly mean temperature	2.14	1.71, 2.78	<0.01	2.34	1.55, 3.9	<0.01
Weekly mean dewpoint	2.08	1.6, 2.81	<0.01	1.52	0.94, 2.5	0.087
Weekly mean relative humidity	0.90	0.85, 0.95	<0.01	1	0.92, 1.11	0.96
Weekly mean barometric pressure	0.55	0.42, 0.71	<0.01	1.42	0.84, 2.19	0.1
Weekly mean PM 2.5 (units)	0.74	0.08, 0.53	<0.01	1.02	0.164, 1.79	0.36
Weekly mean CO (units)	0.23	0.08, 0.53	<0.01	0.57	0.16, 1.79	0.77

† PM 2.5, fine particulate matter; CO, carbon monoxide.

## Data Availability

The dataset is available on request from the authors.

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
