# Peer review of "Clinical, Meteorological, and Air Quality Factors Associated with Ambulatory Pediatric Respiratory Syncytial Virus Infection in Machala, Ecuador, 2018–2023"

_ijerph, 2025, doi:10.3390/ijerph22020272_

Round 1

Reviewer 1 Report

Comments and Suggestions for Authors

In this manuscript, the authors investigated the correlation of RSV activity with meteorological data in Machala, Ecuador from July 2018-July 2023. They revealed the RSV activity was associated with periods of higher mean air temperature and dew points, and lower relative humidity and barometric pressure. PM 2.5 concentrations were positively associated with RSV detection, while CO concentrations were negatively associated with RSV detection events. The finding is useful for determining the ideal timing to improve RSV surveillance and implement RSV prevention measures. It also provided a template for other tropical regions to study the important factors in RSV activity. The manuscript is well-organized and written. There are some concerns before it is published on IJERPH.

Minor concerns,

1.        In the abstract, the abbreviation AUC and the unit mm/week are not easy to understand for readers not in this field.

2.        In the discussion, it will increase the scientific significance if the authors discuss the possible reason why RSV detection was associated with mean outdoor air temperature.

Author Response

  1. In the abstract, the abbreviation AUC and the unit mm/week are not easy to understand for readers not in the field.

Response: On page 1, paragraph 1, lines 19 and 21, in the abstract, the abbreviation AUC has been written out as “area under the curve” and the abbreviation mm/week has been written out as “millimeter/week”. The changes have been made in red in the manuscript file.

  1. In the discussion, it will increase the scientific significance if the authors discuss the possible reason why RSV detection was associated with mean outdoor air temperature.

Response: The following sentence has been added to the manuscript to page 9, lines 273-275 (changes made in red in manuscript): “Stable high temperatures in tropical regions may facilitate RSV survival in large-particle aerosols, particularly during the rainy seasons, allowing for increased transmission during these weather patterns [25].”

Reviewer 2 Report

Comments and Suggestions for Authors

Dear Authors,

Your Original Research Article entitled "Clinical, meteorological, and air quality factors associated with ambulatory pediatric respiratory syncytial virus infection in Machala, Ecuador, 2018-2023." has been carefully reviewed.

This article deserves attention since it highlighted an important topic related to the impact of environmental parameters such as weather, meteorology, air content, precipitation, dew point, and others on the epidemiology of one of the leading causes of acute respiratory tract infection (ARTI) in children aged less than five years old the Respiratory Syncytial Virus (RSV) in a tropical developing country "Ecuador". This paper is very important in helping healthcare providers, and infection prevention and control providers in the implementation of RSV prevention strategies in the mentioned country.

The article is well written in English, well designed, figures and tables are clear for readers.

Here is a list of my remarks, comments, corrections and suggestions related to this paper. Those are divided into Minors and Majors.

Minors:

01- In the Affiliations section, You are kindly invited to put each affiliation on a separate line.

02- In the Keywords section, Kindly remove "RSV" and replace it by "Pediatrics", since you can not put the full name of the virus "Respiratory Syncytial Virus" and its abbreviation "RSV" together in the keywords list.

03- In the Introduction section, When you talked about the importance of studying the meteorological factors related to the epidemiology of RSV, it is good to mention that this point is very important for understanding the epidemiology of different infectious agents during different months, or seasons, in different countries. Many studies covered this point, you can use (not obliged) these articles as references for this point:

--Prevalence, risk factors and seasonal variations of different Enteropathogens in Lebanese hospitalized children with acute gastroenteritis.

--Effects of meteorological factors on influenza transmissibility by virus type/subtype

--Prevalence, antimicrobial resistance and risk factors for campylobacteriosis in Lebanon

--Predicting Salmonella prevalence associated with meteorological factors in pastured poultry farms in southeastern United States

04- In the Materials and Methods section, You are kindly invited to put the reference number for the research approval.

05- In the Lines 140-141, you are kindly invited to replace (Table 1 attached separately) by (Table 1).

06- In the Table 1, You are kindly invited to put the term Symptoms in bold.

07- In the Table 1, Last column, why several parameters such as Nasopharyngitis, Lower Respiratory Tract Infection and Pneumonia, do not show a P-value?

Majors:

01- In your Article you considered that co-infection with RSV and other infectious agent(s) (present in the panel) as member of the group (RSV detected), and you compared different parameters such as Fever, Nasal congestion, cough, wheeze, URTI, LRTI, etc... between the two groups (RSV not detected) V/S (RSV detected), here there is a big issue since several signs and symptoms can be related to other infectious agents in these cases of co-infections, so I suggest to add one more group to your project in which RSV detected group will be as (RSV detected alone) and (RSV detected in co-infections). Noting that out of the 114 cases of RSV detected, 44 are cases of co-infections caused by RSV and one or more other infectious agent. Those 44 case represents 38% of RSV detected cases which represents a high percentage.

02- In the Table 1, column 1, you mentioned that you have patients that are treated with antibiotics, and you mentioned in your Materials and Methods section, that patients treated with antibiotics are excluded from the study, how can you explain this point.

Best Regards.

Author Response

Minor

  1. In the Affiliations section, you are kindly invited to put each affiliation on a separate line.

Response: This change has been made on Page 1, lines 7-11 (changes made in red in the manuscript).

  1. In the Keywords section, kindly remove “RSV” and replace it by “Pediatrics.”

Response: This change has been made on Page 1, Line 25 (changes made in red in the manuscript).

  1. In the Introduction section, when you talked about the importance of studying the meteorological factors related to the epidemiology of RSV, it is good to mention that this point is very important for understanding the epidemiology of different infectious agents during different months or seasons in different countries.

Response: The following sentence has been added to the manuscript at page 2, paragraph 3, lines 55-57: “Understanding epidemiology is important for different infectious agents in the various regions across the world, as has been previously described for both respiratory and gastrointestinal pathogens [20, 21].” Changes are made in red in the manuscript.

  1. In the Materials and Methods section, you are kindly invited to put the reference number for the research approval.

Response: The IRB number 1102402 has been added to page 3, paragraph 1, line 81. Changes are made in red in the manuscript.

  1. In the Lines 140-141, you are kindly invited to replace (Table 1 attached separately) by (Table 1).

Response: This change has been made in the manuscript at page 4, paragraph 2,        line 146 (change made in red).

  1. In Table 1, you are kindly invited to put the terms symptoms in bold.

Response: This change has been made in the manuscript on page 4, in Table 1. Change made in red in the manuscript.

  1. In the Table 1, last column, why several parameters such as nasopharyngitis, lower respiratory tract infection, and pneumonia, do not show a p-value.

Response: The p-values in Table 1 reflect statistical differences between upper and lower respiratory tract infections. Due to low numbers, we did not perform this analysis for each specific diagnosis. This is why there are no p values for nasopharyngitis, laryngotracheitis, bronchiolitis, bronchitis, or pneumonia. The p values are there for the broader diagnoses, including upper respiratory tract infection and lower respiratory tract infection.

Major

  1. In your article, you considered that co-infection with RSV and other infectious agent(s) as a member of the group, RSV detected. You compared different parameters such as symptoms and diagnoses between the two groups. Here there is a big issue since several signs and symptoms can be related to other infectious agents in these cases of co-infections. I suggest to add one more group to your project in which RSV detected group will be as RSV detected, alone, and RSV detected in co-infections.

Response: This change has been made to the manuscript. The additional RSV detected alone and RSV detected in co-infection groups were added to a new table (Table 2). The following sentence was added to the manuscript (changes made in red) on page 5, paragraph 2, line 177-179: “The characteristics of the subjects whose nasopharyngeal sample detected RSV alone and those detecting RSV and another pathogen are shown in Table 2. There were no statistical differences between these two groups.”

  1. In the Table 1, column 1, you mentioned that you have patients that are treated with antibiotics and you mentioned in your Materials and Methods section, that patients treated with antibiotics are excluded from the study. How can you explain this point?

Response: The sentence on page 3, paragraph 1, lines 89-91 were modified in the text (changes in red) to read as follows: “...and those hospitalized or treated with antibiotics within 30 days prior to enrollment were excluded from study participation.”

Round 2

Reviewer 2 Report

Comments and Suggestions for Authors

Dear Authors,

Your revised version of the article entitled "Clinical, meteorological, and air quality factors associated with ambulatory pediatric respiratory syncytial virus infection in Machala, Ecuador, 2018-2023." has been carefully reviewed,

The article is better for publication in the present form, thanks to the modifications you made.

Kindly note that I just want from you to put all the references mentioned in my previous comment number 03.

03- In the Introduction section, When you talked about the importance of studying the meteorological factors related to the epidemiology of RSV, it is good to mention that this point is very important for understanding the epidemiology of different infectious agents during different months, or seasons, in different countries. Many studies covered this point, you can use (not obliged) these articles as references for this point:

--Prevalence, risk factors and seasonal variations of different Enteropathogens in Lebanese hospitalized children with acute gastroenteritis.

--Effects of meteorological factors on influenza transmissibility by virus type/subtype

--Prevalence, antimicrobial resistance and risk factors for campylobacteriosis in Lebanon

--Predicting Salmonella prevalence associated with meteorological factors in pastured poultry farms in southeastern United States.

Best Regards,

Author Response

Comment 1: Kindly note that I just want from you to put all the references mentioned in my previous comment number 03.

03- In the Introduction section, When you talked about the importance of studying the meteorological factors related to the epidemiology of RSV, it is good to mention that this point is very important for understanding the epidemiology of different infectious agents during different months, or seasons, in different countries. Many studies covered this point, you can use (not obliged) these articles as references for this point:

--Prevalence, risk factors and seasonal variations of different Enteropathogens in Lebanese hospitalized children with acute gastroenteritis.

--Effects of meteorological factors on influenza transmissibility by virus type/subtype

--Prevalence, antimicrobial resistance and risk factors for campylobacteriosis in Lebanon

--Predicting Salmonella prevalence associated with meteorological factors in pastured poultry farms in southeastern United States.

Response: These references have been added. Changes made in red in the manuscript, page 2, paragraph 1, line 56.